# OpenReview forum: "Mitigating Sexual Content Generation via Embedding Distortion in Text-conditioned Diffusion Models"
_NeurIPS.cc/2025/Conference — NeurIPS 2025 poster_

### Official Review · Reviewer_eUQG · 2025-06-24

**Clarity:** 2
**Significance:** 4
**Originality:** 3
**Rating:** 5
**Confidence:** 3

**Summary:**

This paper introduces DES (Distorting Embedding Space), a novel defense framework designed to prevent the generation of sexual and NSFW content by text-to-image models. The method focuses on modifying the text encoder, rather than altering the more complex U-Net.

DES works by transforming unsafe text embeddings into safe regions in the embedding space. This is achieved by selecting safe vectors that are minimally similar to unsafe ones and then further subtracting the "nudity" direction to create anti-correlated target vectors. The training process uses three loss components: (1) unsafe loss to push unsafe embeddings away from problematic regions, (2) safe loss to preserve the semantics of safe embeddings, and (3) nudity neutralization loss to eliminate semantic meaning from the “nudity” vector itself, guarding against extraction-based attacks.

Experiments demonstrate that DES significantly reduces attack success rates (ASRs) under various threat models—including explicit prompts, black-box, and white-box attacks—while maintaining high-quality, benign image generation (measured via FID and CLIP score).      Additionally, DES is efficient, requiring only 90 seconds of training and no inference overhead.

**Questions:**

See weaknesses.

**Ethical Concerns:**

["NO or VERY MINOR ethics concerns only"]

**Final Justification:**

The author’s response is convincing and fully addresses my concerns. Therefore, I keep my score of 5.

**Limitations:**

yes

**Quality:**

4

**Strengths And Weaknesses:**

Strengths:

The proposed method demonstrates strong innovation, delivers impressive effectiveness, and operates with low computational cost. The experimental design is extensive, and the analyses are thorough and convincing.

Weaknesses:

1.Dependence on the “nudity” vector: The method’s core mechanism relies on a predefined "nudity" vector. However, the paper does not clearly explain how this vector is obtained, nor does it specify the source of the nudity prompt $p_n$ mentioned in Algorithm 1. The selection of this vector may significantly impact the model's effectiveness.

2. Potential risk of semantic distortion: As DES alters the embedding space, there is a possibility of semantic drift, which may affect the fidelity and accuracy of generated content. The paper does not sufficiently explore or quantify this risk.

3. Methodological complexity: The proposed loss function involves intricate structure with multiple nested operations and adaptive components (e.g., scale factor). This complexity may hinder ease of understanding and reproducibility.

4. Overly complicated presentation in some parts: For instance, Figure 2 (“Overview of DES framework”) is visually dense and difficult to follow. The notation system uses overly similar symbols (e.g., $e_s$, $ê_s$, $ẽ_u$, $ẽ_s$, $ẽ_n$, $e_{neu}$, $ẽ′_s$), which negatively impacts readability and increases cognitive load.

5. Overhead for multi-encoder models: Although DES claims to be efficient, for models like FLUX.1 or SDXL—which use multiple text encoders—training each encoder independently may incur significant cumulative cost in practice. This trade-off is not sufficiently addressed.

---

> ### Author Rebuttal · Authors · 2025-07-31
>
> We appreciate for the positive assessment and insightful feedback. We have addressed the identified weaknesses below.
>
> - **Dependence on the “nudity” vector**
>
> There are several common concepts used to evaluate the performance of concept removal methods. When targeting sexual content, 'nudity' is a widely-used concept [4, 5]. Our work follows this practice.
>
> [4] Yimeng Zhang, Xin Chen, Jinghan Jia, Yihua Zhang, Chongyu Fan, Jiancheng Liu, Mingyi Hong, Ke Ding, and Sijia Liu. Defensive unlearning with adversarial training for robust concept erasure in diffusion models. In Advances in Neural Information Processing Systems (NeurIPS), 2024.
>
> [5] Rohit Gandikota, JoannaMaterzynska, Jaden Fiotto-Kaufman, and David Bau. Erasing concepts from diffusion models. In Proceedings of the IEEE/CVF International Conference on Computer Vision, pages 2426–2436, 2023.
>
> - **Potential risk of semantic distortion**
>
> We attempted to minimize the semantic drift of safe data, while maximizing the semantic drift of unsafe data. To achieve it, we utilized the safe loss $\mathcal{L}_s$ in addition to the unsafe loss $\mathcal{L}_u$. Optimizing only $\mathcal{L}_u$ leads to significant semantic drift on safe samples, however, jointly optimizing $\mathcal{L}_u$ and $\mathcal{L}_s$ can mitigate their semantic drift.
>
> - **Methodological complexity**
>
> We will release our source code as open-source to facilitate understanding and reproducibility.
>
> - **Overly complicated presentation in some parts**
>
> Thank you for the feedback on readability of our paper. We will further improve figures and notations for clarity. For example, converting target vectors $\hat{e}\_{s,i}$ to $t\_i$, unsafe vectors $e\_{u,i}$ to $u\_i$, safe vectors $e\_{s,i}$ to $s\_i$, nudity-integrated current safe vectors $\tilde{e}'\_{s,i}$ to $\tilde{s}^n\_i$, nudity vector $e\_n$ to $n$, and neutral vector $e\_{neu}$ to $e\_{\phi}$. We believe these modifications will enhance readability.
>
> - **Overhead for multi-encoder models**
>
> This is a fair point regarding the practical overhead for multi-encoder models. While DES is highly efficient on a per-encoder basis (requiring only 90 seconds for a single CLIP encoder), we acknowledge that the cumulative training cost is an important consideration for models that employ multiple text encoders, such as FLUX.1 or SDv3.x. This is a limitation of the current approach, and we agree that developing more integrated or cost-effective strategies for multi-encoder architectures is an important direction for future work.

---

> > ### Comment · Reviewer_eUQG · 2025-08-01
> >
> > Thank you for providing a clear and satisfactory rebuttal on the raised questions, including the “nudity” vector, handling of semantic drift, improvements to presentation, and acknowledgment of limitations for multi-encoder models. I find the responses convincing and believe the rebuttal adequately addresses my questions.

---

> > > ### Author Response · Authors · 2025-08-01
> > >
> > > Thank you so much for taking the time to read our paper and the responses.

---

### Official Review · Reviewer_KmLy · 2025-07-01

**Clarity:** 3
**Significance:** 3
**Originality:** 3
**Rating:** 4
**Confidence:** 4

**Summary:**

The paper proposes Distorting Embedding Space (DES) as a defense mechanism against sexual content generations with diffusion models. Unlike existing paradigm that incurs additional inference costs or leads to degradation in image quality, the proposed method is designed to remove unsafe concepts while preserving the image quality with zero additional inference costs. To do so, it transforms unsafe embeddings in the text encoder to safe embedding regions and also neutralizes "nudity" embeddings to empty concepts. The proposed methods achieves SoTA performance at preventing generation of harmful concepts while maintaining benign image quality.

**Questions:**

N/A

**Ethical Concerns:**

["NO or VERY MINOR ethics concerns only"]

**Final Justification:**

The authors have addressed my concerns, so I keep my initial rating of Borderline Accept. I encourage the authors to explore my concern on expressing only a single type of harmful concept with a single vector in future works.

**Limitations:**

Yes.

**Quality:**

3

**Strengths And Weaknesses:**

Strengths:
1. The paper introduces a defense method against sexual content generations on diffusion models that leads to zero additional cost during inference time.
2. The empirical results show SoTA robustness performance while maintaining image fidelity.
3. The paper designs the "harmful" and "safe" vectors designed to guide harmful embeddings towards safe embeddings.

Weaknesses:
1. While shown by the results in Table 5, is the proposed method generalizable to various harmful concepts? Harmful concepts like nudity, violence, horror, and hate are semantically different and unlikely to be linearly separable by a single "harmful" direction. The method assumes that subtracting a single “nudity” vector (or its analog for other concepts) is sufficient to remove the harmful attribute, but this overgeneralizes the structure of the embedding space. In reality, I assume that different harmful concepts should be assigned different "safe" vectors.
2. While the image quality is maintained in Table 1, it is degraded more in Table 5. Why is this so?
3. The embedding space of the text encoder is high dimensional and not linear. Thus, representing the "harmful" and "safe" concepts using a linear vector may be oversimplifying the underlying representations. How can the authors ensure that these vectors are optimal?

---

> ### Author Rebuttal · Authors · 2025-07-31
>
> We thank the reviewer for the insightful comments on our work. We acknowledge the strengthes and weaknesses you have pointed out and offer the following clarifications.
>
> - **Single Nudity Vector Issue**
>
> We agree that devising a method based on multiple harmful directions could be an effective way to improve overall performance, and we appreciate you raising this insightful point. It is worth noting that our current approach, which uses a single vector "nudity" for the single-concept experiment and "nudity, blood, politics" for the multi-concept experiment, is highly effective. Nevertheless, we appreciate for a valuable suggestion and consider it a promising area for future research to elaborate on.
>
> - **Image Quality Issue**
>
> That is a good point. The degree of image quality degradation can vary depending on the specific concept being unlearned. This is a common issue for concept removal-based approaches. For example, AdvUnlearn [4] showed this effect, with FID scores of 19.34, 16.96, and 18.06 for the nudity, Van Gogh, and church concepts, respectively.
>
> [4] Yimeng Zhang, Xin Chen, Jinghan Jia, Yihua Zhang, Chongyu Fan, Jiancheng Liu, Mingyi Hong, Ke Ding, and Sijia Liu. Defensive unlearning with adversarial training for robust concept erasure in diffusion models. In Advances in Neural Information Processing Systems (NeurIPS), 2024.
>
> - **Optimal Vector Issue**
>
> Our method was devised from the empirical observation that enforcing a strong negative correlation between a “nudity” vector and designated safe vectors reliably suppresses the nudity concept in the text embedding. It is considered to be linear, but it proves effective at both mitigating sexual content and maintaining image quality. Importantly, some works, such as negative prompt and Safe Latent Diffusion, already indicate that subtracting directions associated with harmful concepts can yield safer generations. Those methods operate in the U-Net latent space, which is also high-dimensional and non-linear, yet direction subtraction remains effective. This supports the plausibility of using an unsafe vector in the CLIP embedding space, which is high-dimensional and non-linear, as well. Nevertheless, we do not expect that it is optimal. In future work, we plan to investigate more complex techniques, such as decomposing unsafe semantics from vectors, to enable more targeted suppression while further preserving benign attributes.

---

> > ### Comment · Reviewer_KmLy · 2025-08-05
> > **Thank you for the rebuttal**
> >
> > I appreciate the authors for providing the rebuttal. My concerns are addressed, and I keep my initial rating of Borderline Accept.

---

> > > ### Author Response · Authors · 2025-08-05
> > >
> > > We are pleased that our rebuttal addressed your concerns and appreciate your positive assessment.

---

### Official Review · Reviewer_EXAa · 2025-07-03

**Clarity:** 3
**Significance:** 3
**Originality:** 3
**Rating:** 4
**Confidence:** 4

**Summary:**

The paper proposes a method to prevent the generation of sexual contents in T2I models such as FLUX and stable diffusion combining three techniques to alter the text encoder only. First, DES aligns "unsafe" embeddings with target safe vectors where the target is determined as the least similar vector to the given unsafe embedding among safe embeddings. They further enhance this by subtracting the direction of the nudity vector. Second, because the said alignment process can mess up the internal representations of the text encoder, they enforce a form of a regularization loss to preserve original safe embeddings scaled by the distance from the nudity embedding. Finally, they "neutralize" the nudity embedding by aligning it with the null embedding (the empty string). The method results in the state-of-the-art attack success rate and outperforms baseline methods while preserving reasonable degree of original generation quality.

**Questions:**

- Do you have a more specific explanation as to why the effectiveness in the case of FLUX.1 is worse? Is it CLIP that contributes more or the T5 encoder, or perhaps the fact that you have to optimize both at the same time?
- Are CLIP score and FID truly enough to reliably measure perceived image generation quality? A human study would be necessary better confirm the generation quality.
- Does the scaling factor vary significantly across different concepts?
- Does preventing multiple concepts simultaneously degrade generation quality even further? I am curious as to what exactly happens in the optimization process if you use the vector that corresponds to the text string "nudity, blood, politics". Is it similar to the average of the three embeddings computed separately or is it something entirely different?

**Ethical Concerns:**

["NO or VERY MINOR ethics concerns only"]

**Final Justification:**

The work proposes a well-working solution to a well-motivated problem, and the author rebuttal addressed most of my previous concerns. Therefore, I think this work can be a useful addition to the community, and keep my score of 4.

**Limitations:**

yes

**Quality:**

3

**Strengths And Weaknesses:**

**Strengths**
- The method seems to rather exhaustively prevent diverse loopholes potentially overseen by prior works based on the observation that the safe embeddings may be aligned with the nudity vector. This might be one of the main reasons why the given method outperforms the baseline methods.
- The qualitative and quantitative results demonstrate that the method achieves great generation quality as well as attack success rates, outperforming most baseline methods. This holds for more advanced methods such as adversarial prompts or even white-box methods such as UDA, which is impressive.
- Extensive experiments are done to show the effectiveness of the method, including I2I tasks.
- Embedding space analysis in Sec. 5 is interesting, and the fact that adversarial prompts are aligned with the nudity vector might be a useful resource for future works.

**Weaknesses**
- Although the attack success rate is great, the image quality measured by CLIP score seems to be somewhat lower compared to other baseline methods (Fig. 18), begging a question as to whether there is a quality-prevention tradeoff. The proposed method will be more useful if the preservation of generation quality was even better.
- The method, although still better than baseline methods, is less effective in the case of FLUX.1 which uses both CLIP and T5. As future generation models might shift towards ensembled text encoders, this weakness might need to be handled for real-world applicability.
- The design of the method seems to rely on specific observations floating around singular and specific text embeddings such as "nudity" (e.g., observation that safe embeddings exhibit correlations with the nudity vector), and this might limit generalization to more challenging cases such as defending against multiple concepts, or might require per-concept fine-tuning (i.e., the optimal scaling parameter in Eq. (4) might be different across different concepts).

---

> ### Author Rebuttal · Authors · 2025-07-31
>
> We sincerely thank you for your thoughtful and constructive review. We have addressed your questions below.
>
> - **FLUX.1 Issue**
>
> We trained CLIP and T5 encoders independently to handle scenarios involving ensembled text encoders. We acknowledge that our method's effectiveness was less pronounced on FLUX.1 compared to single-encoder models. We believe this is because the multi-encoder setting is inherently more challenging to defend. An attacker can leverage and combine the vulnerabilities of each distinct text encoder, making it more difficult for a defense mechanism to be robust across all of them simultaneously. This remains an open problem, as robust solutions for defending multi-encoder models have yet to be established. We agree that developing defenses specifically tailored for these complex, ensembled models is a crucial direction for future research.
>
> - **Metric Issue**
>
> We agree that while CLIP score and FID are standard automated metrics in the field, they do not perfectly capture the nuances of perceived image quality from a human perspective. Consequently, several studies for generative models developed evaluation metrics that better reflect human perception. To that end, we expanded our evaluation to include advanced metrics like BLIPScore [1], PickScore [2], ImageReward [3]. Our results, shown in Table 10 (Section B.8), demonstrate that DES maintains good performance under these evaluations. Specifically, DES achieved a PickScore of 21.02 and an ImageReward of -0.032, comparing to the AdvUnlearn, which recorded scores of 20.73 and -0.622, respectively.
>
> [1] Junnan Li, Dongxu Li, Silvio Savarese, and Steven Hoi. Blip-2: Bootstrapping language-image pre-training with frozen image encoders and large language models. In International conference on machine learning, pages 19730–19742. PMLR, 2023.
>
> [2] Yuval Kirstain, Adam Polyak, Uriel Singer, Shahbuland Matiana, Joe Penna, and Omer Levy. Pick-a-pic: An open dataset of user preferences for text-to-image generation. Advances in Neural Information Processing Systems, 36:36652–36663, 2023.
>
> [3] Jiazheng Xu, Xiao Liu, Yuchen Wu, Yuxuan Tong, Qinkai Li, Ming Ding, Jie Tang, and Yuxiao Dong. Imagereward: Learning and evaluating human preferences for text-to-image generation. Advances in Neural Information Processing Systems, 36:15903–15935, 2023.
>
> - **Scaling Factor Issue**
>
> In our paper, the scaling factor $s_g$​ was first determined for the nudity concept, as detailed in Table 15, and this value was subsequently applied to experiments involving other concepts. In this rebuttal, we have conducted additional experiments on $s_g$ for other NSFW concepts and Van Gogh concept. Our experiments suggest that the most effective $s_g$ remains within a relatively close range, as shown in the tables below. For example, for other NSFW concepts, the best performance was observed when $s_g \in \{200, 250\}$. For the Van Gogh concept, the range yielding the best results was slightly broader at $s_g \in \{150, 200, 250\}$. This indicates that the value of $s_g$ does not vary significantly according to the target concept.
>
> **[Impact of scaling factors ($s_g$) on other NSFW concepts.]**
>
> | $s_g$ | I2P-Violence↓ | I2P-Illegal↓ | I2P-Hate↓ | I2P-Selfharm↓ | I2P-Harassment↓ | I2P-Shocking↓ | Avg. ASR↓ | CLIP score↑ | FID↓ |
> | :---: | :---: | :---: | :---: | :---: | :---: | :---: | :---: | :---: | :---: |
> | 0 | 31.35 | 11.14 | 12.55 | 26.47 | 15.29 | 33.18 | 21.66 | 25.63 | 16.96 |
> | 50 | 16.40 | 5.36 | 6.93 | 9.86 | 8.01 | 15.07 | 10.27 | 25.57 | 17.15 |
> | 100 | 9.79 | 3.30 | 3.03 | 4.24 | 4.37 | 10.40 | 5.86 | 25.49 | 17.64 |
> | 150 | 6.22 | 1.65 | 2.16 | 2.00 | 4.00 | 6.54 | 3.76 | 25.14 | 18.35 |
> | 200 | 4.23 | 1.10 | 0.87 | 0.50 | 1.33 | 3.27 | 1.88 | 24.90 | 19.10 |
> | 250 | 3.31 | 1.10 | 0.43 | 0.87 | 1.46 | 3.27 | 1.74 | 24.41 | 19.86 |
> | 300 | 1.98 | 0.28 | 0.00 | 0.50 | 1.21 | 1.87 | 0.97 | 22.73 | 25.55 |
>
> **[Impact of scaling factors ($s_g$) on Van Gogh concept.]**
>
> | $s_g$ | ASR↓ | CLIP score↑ | FID↓ |
> | :---: | :---: | :---: | :---: |
> | 0 | 12.0 | 25.99 | 17.15 |
> | 50 | 6.0 | 26.08 | 16.91 |
> | 100 | 6.0 | 26.11 | 16.70 |
> | 150 | 2.0 | 26.06 | 16.75 |
> | 200 | 2.0 | 26.08 | 16.67 |
> | 250 | 2.0 | 26.05 | 16.64 |
> | 300 | 4.0 | 26.03 | 16.64 |
>
> - **Multi Concept Issue**
>
> We appreciate your insightful question regarding the model's behavior in a multi-concept scenario. First, we observed that there is a slight degradation in image quality when preventing multiple concepts simultaneously, compared to preventing a single concept. Second, the embedding for a text sequence like "nudity, blood, politics" is not identical to the average of the individual concept embeddings. This is due to the contextual nature of the text encoder, where interactions between tokens influence the final embedding. To quantify this, we calculated the cosine similarity between the averaged embedding and the embeddings of different permutations of the phrase. As shown in the table below, the cosine similarities are ranged 0.746~0.784, confirming they are close but meaningfully different.
>
> | Text Sequence | Cosine Similarity with Average embedding |
> | :---: | :---: |
> | "nudity, blood, politics" | 0.746 |
> | "nudity, politics, blood" | 0.749 |
> | "politics, nudity, blood" | 0.755 |
> | "politics, blood, nudity" | 0.763 |
> | "blood, nudity, politics" | 0.763 |
> | "blood, politics, nudity" | 0.784 |

---

> > ### Comment · Reviewer_EXAa · 2025-08-09
> >
> > I appreciate the detailed response by the authors. The multi concept part is interesting and sensible, and I think it is worth considering this as future research direction. The scaling factor ablation could be added in the final draft of the work. Overall, most of my concerns have been addressed. Therefore, I keep my rating of 4.

---

> > > ### Author Response · Authors · 2025-08-09
> > >
> > > Thank you for the review and comments. As suggested, we will add a scaling-factor ablation in the final version. We appreciate that most of your concerns have been addressed and will incorporate these updates to strengthen the paper.

---

### Note · Authors · 2025-08-13

Dear Chairs and Reviewers,

We thank the chairs and the reviewers for their time and feedback. We're pleased that all reviewers consistently responded positively to our contributions.

Our proposed method, DES, effectively mitigates sexual content generation through the novel embedding space control. It shows state-of-the-art (SOTA) performance in mitigating sexual content generation on diverse models, including recent image generation model, such as FLUX.1 [1] and Stable Diffusion v3.5 [2]. It is not limited to the text-to-image task, but can be extended to other tasks like image-to-image. Moreover, DES is highly efficient to train and requires no additional inference overhead. Extensive experiments demonstrate that DES consistently outperforms previous SOTA methods across different models and tasks. For example, DES achieves an ASR of 9.47% on FLUX.1, which is a 76.5% improvement over EraseAnything [3], and 0.52% on Stable Diffusion v1.5 [4], which is a 63.9% improvement over AdvUnlearn [5].

Furthermore, the reviewers acknowledged the significance of our study, such as:
- Our method is constructed from empirical observations that may have been overlooked by previous approaches.
- Extensive experiments demonstrate its strong effectiveness with low computational cost.
- Our analysis of the embedding space is convincing and useful for future works.
- Our innovative ideas to guide unsafe embeddings toward safe regions to prevent unsafe content generation.

If our paper is accepted, we will reflect the feedback from the chairs and the reviewers in the camera-ready version. We hope that the final decision will be made with thoughtful consideration of these points.

Thank you again for your efforts and time.

Best regards,
The authors

[1] Labs, Black Forest, et al. "FLUX. 1 Kontext: Flow Matching for In-Context Image Generation and Editing in Latent Space." arXiv preprint arXiv:2506.15742 (2025).
[2] Esser, Patrick, et al. "Scaling rectified flow transformers for high-resolution image synthesis." ICML 2024.
[3] Gao, Daiheng, et al. "Eraseanything: Enabling concept erasure in rectified flow transformers." ICML 2025.
[4] Rombach, Robin, et al. "High-resolution image synthesis with latent diffusion models." CVPR 2022.
[5] Zhang, Yimeng, et al. "Defensive unlearning with adversarial training for robust concept erasure in diffusion models." NeurIPS 2024.

---

### Decision · Program_Chairs · 2025-09-17

**Decision:**

Accept (poster)

**Comment:**

This submission proposes a new method for addressing the important problem of sexual content generation in T2I diffuson model. The algorithm modifies “unsafe” embeddings with “neutral” content embeddings that are determined as being the most dissimilar to the set of unsafe embeddings. For more robustness the “nudity” vector is removed as well. The second component of this algorithm for utility preservation is a standard formulation of adding a regularization term to the overall objective. The final component is again a similar component from prior works on aligning “nudity” to the “null” embedding.

The strength of this paper is in the “defense-in-depth” style strategy of combining many different existing components from the “controllability” and “unlearning” literature with some modifications tailored to sexual content generation. This in of itself leads to a new algorithm that shows great results. Additionally, the experimentation is thorough and covers many different scenarios where sexual content generation can occur. Additionally, I believe the focus on sexual content generation solely is a strength of the paper as this issue requires great care in addressing and has its own challenges that can be separate from the mitigation from the other content types such as copyright.

The weaknesses of the paper, most of which were addressed in the rebuttal included whether the current evaluation was rigorously assessing image quality, the choice to assume linearity of the representation space, and issues of overhead due to needing to modify multiple text encoders in certain models like FLUX. The latter remains after the rebuttal but does not detract from the utility of this work. One concern that was brought up by reviewers that I encourage the authors to consider is which of the observations the method was built on are specific to sexual generation and which ones can generalize to other settings. I think this would help the broader “unlearning” and “safety” in T2I model community gain more from the paper. This currently remains as a question / concern amongst most reviewers.

Overall, I think this work addresses an important problem and provides important insights on how to address unwanted content generation. I think the choice to focus on sexual content generation and to use this choice to tailor the method is valuable. It raises the question of whether we should be aiming to build “general” mitigation / concept unlearning methods or if a more “domain tailored” approach is necessary. Thus I am recommending acceptance.